# A Possible New Diagnostic Method for Early Diagnosis of Cryptococcus Infection in Lymphoma Patient Co-Infected with SARS-CoV-2

**DOI:** 10.3390/reports6010011

**Published:** 2023-03-01

**Authors:** Manuela Colosimo, Salvatore Nisticò, Francesco Quintieri, Annamaria De Luca, Pasquale Minchella, Luca Gallelli

**Affiliations:** 1Microbiology and Virology Unit, Pugliese-Ciaccio Hospital, 88100 Catanzaro, Italy; 2Operative Unit of Infectious Disease, Pugliese-Ciaccio Hospital, 881000 Catanzaro, Italy; 3Operative Unit of Clinical Pharmacology and Pharmacovigilance, Department of Health Science, University of Catanzaro, 88100 Catanzaro, Italy

**Keywords:** Cryptococcus, dilution, screening, treatment

## Abstract

A Cryptococcus subspecies, *neoformans*, represents the most pathogenic infection for humans, particularly in immunocompromised hosts (e.g., cancer patients, drug users). In the present study, we described a 67-year old woman with non-Hodgkin lymphoma who developed an infectious disease sustained by *Cryptococcus neoformans.* Biochemical data documented a decrease in lymphocytes count while clinical evaluation was suggestive on meningeal infection. The microbiological analysis of the serum, using a dilution pattern through the CrAg lateral flow assay (Immy, Norman, OK 73069, USA) detected the antigen of *Cryptococcus* (dilution 1/1280), and a treatment with liposomal amphotericin B (3 mg/kg id) plus flucytosine (100 mg/kg per day orally in four divided doses) were started, showing an improvement of symptoms. This case report suggests that an antigen dilution can be used to perform a rapid diagnosis and to quickly start the pharmacological treatment.

## 1. Introduction

Cryptococcus is free-living encapsulated saprophytic yeast. Human infections are caused by *Cryptococcus neoformans* and *cryptococcus gattii*. The epidemiology and the clinical and molecular characteristics of these two species vary. C. neoformans is classified into C. neoformans var. grubii (serotype A) and C. neoformans var. neoformans (serotype D) [1].

Among the many species belonging to the genus *Cryptococcus*, the subspecies *neoformans* represents the most pathogenic infection for humans, particularly in immunocompromised hosts [2].

*C. neoformans* is a ubiquitous and opportunistic fungal pathogen encapsulated (30 μm thick) unicellular yeast, is about 5–12 μm in diameter, and is primarily responsible for cryptococcal meningitis [3]. The colonies are white to creamy and subsequently take on a sticky appearance.

Several studies suggest that impaired host cell-mediated immunity such as human immunodeficiency virus (HIV) infection autoimmune diseases, immunosuppressive therapy (e.g., corticosteroids, anticancer drugs), liver cirrhosis, lung diseases, lymphoproliferative malignancy and hematological malignancy, play a central role in the pathogenesis of cryptococcal infections [4,5,6,7]. In the present study, we describe a woman with non-Hodgkin lymphoma that developed a meningeal infection sustained by *C. neoformans.*

## 2. Case Presentation

A 67-year-old Caucasian woman, with an history of non-Hodgkin Lymphoma and previous positivity to COVID-19, presented to our ward on 14 May 2021, due to symptoms of 3 days of evolution of occipital headache that woke her up at night, fever (38 °C) and temporo-spatial disorientation. On exploration, she was hemodynamically stable, without respiratory dysfunctions, but with a disorientation in time, place, and person. She presented a stiff neck and Kernig and Brudzinski signs were positive. Biochemical analysis revealed a severe decrease in lymphocytes and platelet count, and an increase in C Reactive protein and neutrophils count (Table 1). A total body computed tomography (CT) scan with contrast showed a large (2.35 cm) hypodense cerebrospinal fluid-like lesion affecting the cerebral and cerebellar parenchyma, without associated edema (Figure 1), that was not suggestive of meningeal infectious disease.

Considering clinical and laboratory data as well as the history of immunosuppression, the diagnosis of meningitis probably related to *Cryptococcus* infection was postulated and the patient was hospitalized.

Considering a possible fungal infection in immunosuppressed patients, Β-d glucan was requested, but it was referred as negative.

Blood samples were sent to microbiology operative unit for the analysis. In in blood samples of this woman, using a dilution pattern through the CrAg lateral flow assay (Immy, Norman, OK 73069, USA), we detected the antigen of *Cryptococcus* (dilution 1/1280). It is a dipstick sandwich immunochromatographic assay used for the qualitative or semi-quantitative detection of the capsular polysaccharide antigens of Cryptococcus species complex (*Cryptococcus neoformans* and *Cryptococcus gattii*), and the treatment with liposomal amphotericin B (3 mg/kg id) was started. The lumbar puncture was not performed because it was considered a high-risk procedure and the patient did not agree.

Two days later, the serological diagnosis was confirmed by broth microdilution plates of blood cultures coming back positive for *C. neoformans* with a high sensitivity to Amphotericin B (minimum inhibitory concentration, MIC: 0.5 μg/mL), Posaconazole (MIC: 0.25 µg/mL), Voriconazole (MIC: 0.25 µg/mL), and Itraconazole (MIC: 0.125 µg/mL) (Figure 2). The detection of *C. neoformans* was performed using the Maldi toff (Becton Dickinson, Four Oaks, NC 27524, USA) system, the automatic system for microbial identification using mass spectrometry (Figure 2), while the VitekS2 and out the Micronaut-AM system (Merlin broth test in microdilution system) was used for the sensitivity test (Figure 3 and Table 2).

Clinical evaluation revealed an improvement of clinical condition and, about 14 days later (June 3), a new evaluation of antigen, CrAg lateral flow assay (Immy, Norman, OK 73069, USA), revealed the persistence of the high value of dilution (1/1280), therefore, Flucytosine (100 mg/kg per day orally in four divided doses) was added to Amphotericin with a progressive improvement of clinical conditions and the patient was discharged after 8 days (11 June) on Fluconazole (400 mg per day). One month later, during the follow-up the patient does not show signs or symptoms of meningitis and the antigen in serum was not detected. Patients maintained close hematological surveillance under regular appointments at the Onco-Haematology Clinic of the same hospital.

## 3. Discussion and Conclusions

We reported a meningeal infection sustained by *Cryptococcus neoformans* in a woman with non-Hodgkin lymphoma and COVID-19.

Several authors described the development of meningitis sustained by *C. neoformans* in patients with immunocompromising conditions [7].

The Centers for Disease Control and Prevention USA documented in an epidemiological study that worldwide, there are almost one million new cryptococcal meningitis cases reported annually, suggesting that this infection represents an important global health concern.

In particular, it has been reported that patients with lymphoproliferative disorders, including malignant lymphoma, have a cell-mediated immunity suppression with intercurrent cryptococcal infections complicated with Acute Lymphocytic Leukemia, Adult T-cell Leukemia, Hairly Cell Leukemia, and Chronic Lymphocytic Leukemia [8,9,10,11].

In an Italian 10-year retrospective study, Pagano et al. [12] described 17 cases of cryptococcosis (8 with acute leukemia and 3 with non-Hodgkin lymphoma) in patients with hematological malignancies. More recently, Reisfeld-Zadok et al. [11] reported the development of Cryptococcal meningitis infection in 2 patients with chronic lymphocytic leukemia.

Hirai et al. [13], reviewed literature data for malignant lymphoma complicated by cryptococcal infection, reporting that 13 of 17 cases (76%) had a disseminated infection (e.g., meningitis or fungemia or intramuscular abscess), with a mortality of 41% in all cases and 54% in disseminated cases. The authors documented that several factors contributed to the etiology of disseminated cryptococcosis (e.g., non-Hodgkin’s lymphoma; cellular immunodeficiency associated with chemotherapy; cyclic hematopoietic disorders due to chemotherapeutic regimens including rituximab and corticosteroids).

Finally, Zhang et al. [14] described a 20-year-old male patient affected by Hodgkin lymphoma that developed eosinophilic meningitis that was probably related to cryptococcus infection responsive to antifungal therapy.

However, very few data have been published on the topic of non-Hodgkin and cryptococcus infection. In fact, when searching on PubMed “cryptococcus” and “lymphoma” there are 205 results, while searching “cryptococcus” and “non-Hodgkin lymphoma” shows that there have been 68 published papers. Of these, the most common infection involves the respiratory system, and only 10 reports (from 1958 to 2022) describe a cerebral infection sustained by cryptococcus in patients with non-Hodgkin lymphoma. In these reports, patients received a treatment with chemotherapy regimen [13,15] or after autologous stem cell transplant [16] or after monoclonal antibody treatment [17]. In contrast in our case, the patient has a history of non-Hodgkin lymphoma in follow-up and without chemotherapy treatment. However, in agreement to other studies [8,9,10,11], in our patient biochemical values documented a severe decrease in lymphocytes count and an increase in neutrophils one.

In a previous study, Korfel et al. [9], documented the development of a systemic mycosis caused by Cryptococcus species in two patients affected by Hodgkin’s lymphoma stage IVB (Ann Arbor). In this study, the authors cited 54 cases of cryptococcosis in patients with an history of Hodgkin’s lymphoma; in these patients, laboratory findings documented an absolute lymphopenia that probably played a role in the development systemic infectious disease.

In our case, the decrease of lymphocytes count could be related to the treatment with glucocorticoids for COVID-19, in agreement with current treatments [18]. In fact, in a case series of cryptococcal infections presented by Memorial Sloan-Kettering Cancer Center, it was reported that 38 of 46 patients infected by cryptococcus (from 1956 to 1972) were affected by leukemias or lymphomas, while 39 of these were treated with glucocorticoids [19]. In this group of patients, cryptococcal meningitis was commonly diagnosed. Moreover, all patients with cryptococcal infection were found to have lymphopenia. Similarly, a case series conducted from 1989 to 1999 at M D Anderson Cancer Center documented the presence of cryptococcal infection in 31 cancer patients (20 of whom with hematologic cancers); about 61% of these patients had lymphopenia and more than half (52%) were treated with glucocorticoids [20]. In order to reduce the time of diagnosis, we used the antigen dilution. Even if MALDI-TOF is necessary for an appropriate diagnosis of the species of pathogens, the antigen dilution can be used to perform a rapid diagnosis of the pathogen involved in the development of infection, so as to quickly start the pharmacological treatment.

To date there are three categories of methods that can be used to diagnose cryptococcal meningitis: India Ink microscopy, which can be used on cerebrospinal fluid (CSF); culture, which can be used on CSF or blood; and antigen detection.

Microscopy is a fundamental, easy-to-use technique but its sensitivity is dependent on the quality of the specimen and the experience of the laboratory personnel. However, microscopy has another limitation related to the stains used [21]. In fact, the India link is a commonly used stain and has a very low sensitivity (86%) that is decreased to 42% among persons with cerebrospinal fluid cultures with <1000 colony-forming units/mL [21].

In contrast, there are several methods to detect cryptococcal antigen in CSF or serum: latex agglutination (LA), enzyme immunoassay (EIA), and lateral flow assay (LFA) (Table 3).

Lateral agglutination is the first test used in the clinical course for the diagnosis of fungal infection [1]. This technique uses antibodies raised in rabbits against whole cryptococcal cells and passively coated onto latex beads. This assay detected glucuronoxylomannan, the major capsular polysaccharide of *C. neoformans,* that is shed in large amounts into the blood and CSF during the cryptococcal meningitis infection [22].

The enzyme immunoassay is an automated spectrophotometric method commonly used in the management of biomarkers [23]

LFA is a semiquantitative test that can be used to measure disease burden by determining the titers for positive results [23]. This test uses gold-conjugated, monoclonal antibodies impregnated onto an immunochromatographic test strip to detect cryptococcal capsular polysaccharide glucuronoxylomannan antigen for all four C. neoformans serotypes (A–D) [24]. If cryptococcal antigen is present in a specimen, suspended, gold-conjugated antibodies bind to the antigen.

In a previous study Panackal et al. [25], the authors compared the sensitivity of the EIA and the latex agglutination test on 185 blood and 164 cerebrospinal fluid (CSF) samples obtained from patients with cryptococcosis without known immunocompromising conditions. In this study, the authors documented that the LA assay was more sensitive than the EIA [25].

Comparing these tests, we evaluated that the advantage of the LFA method is that it is simple to use, and the results are available in 10 min, with high sensitivity (95%) and low costs (about EUR 4) for each test.

In this infectious condition the time of the treatment is important to reducing the risk of severe disease and death. In fact, in this case after the clinical evaluation and the antigen test, a treatment with liposomal Amphotericin B improved clinical symptoms.

In conclusion our case supports the use of an antigen test (lateral flow assay) in cryptococcus infection in patients at high risk (cancer disease, chemotherapy, glucocorticoid treatment) to obtain a rapid diagnosis and to start a rapid treatment.

## Figures and Tables

**Figure 1 reports-06-00011-f001:**
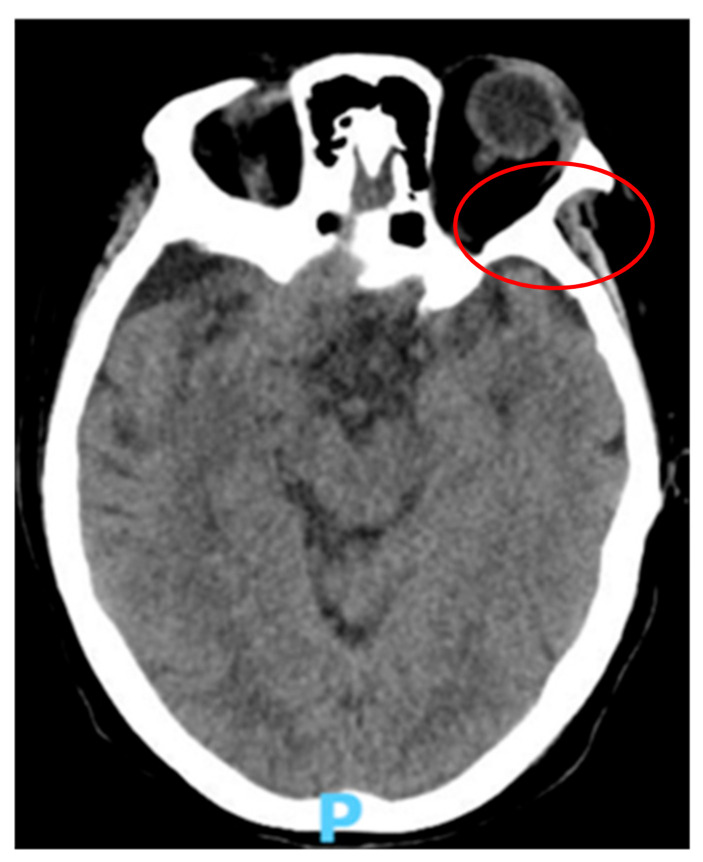
Head computed tomography (CT) scan revealing a large (2.35 cm) hypodense cerebrospinal fluid-like lesion (red circle).

**Figure 2 reports-06-00011-f002:**
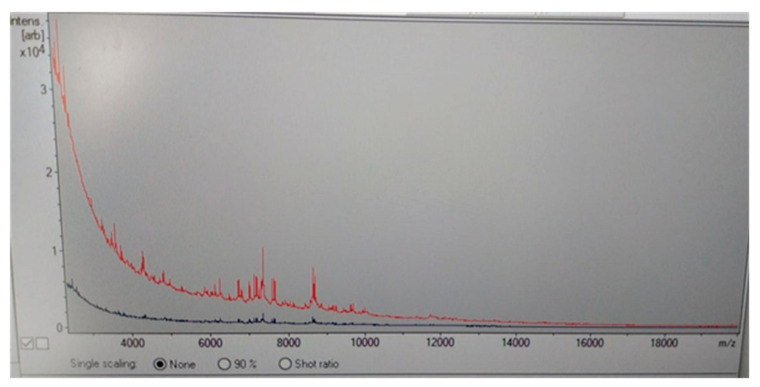
Maldi Toff spectrometry of Cryptococcus neoformans colony (red line) respect to control line (blue line).

**Figure 3 reports-06-00011-f003:**
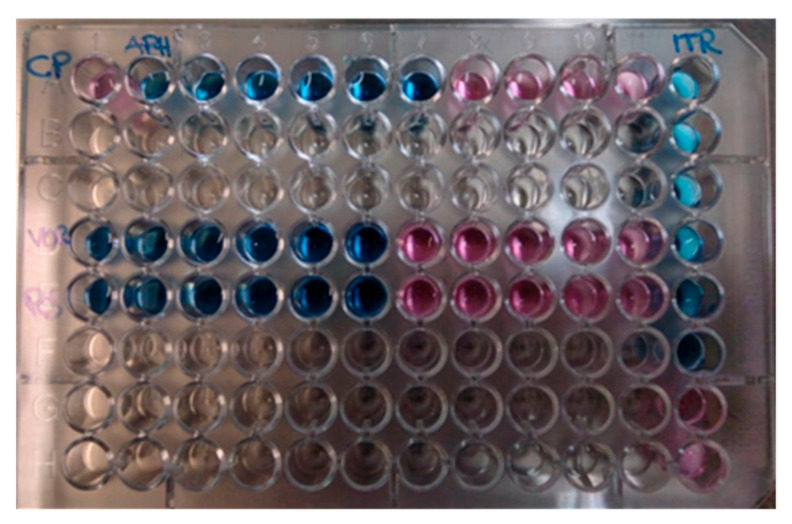
Broth microdilution plates of blood cultures. In blue: sensible concentration. In red: resistant concentration.

**Table 1 reports-06-00011-t001:** Biochemical findings in enrolled woman, during the hospitalization.

	Normal Range	14 May	20 May	23 May	30 May	3 June	6 June	9 June
Blood Cells	3.92–5.13 million cells/µL	4.8	4.9	4.9	6.5	4.3	3.9	4.5
White Cells	3.4–9.6 billion cells/L	4.6	4.5	4.4	4.2	3.8	3.4	3.4
Hemoglobin	11.6–15 g/dL	13.2	12.6	12.1	14.4	12.2	9.2	9.3
Platelets	157,000–371,000/mcL	18,900	175,000	192,000	123,000	103,000	98,000	91,000
Neutrophils	40–60%	78.1	77.5	77.1	82.5	77.5	78.6	75.4
Lymphocytes	20–40%	14.1	14.1	13.9	9.9	9.8	13.0	17.5
Monocytes	2–8%	5.2	6.0	6.6	4.5	5.4	4.8	3.2
Eosinophils	1–4%	0.0	0.1	0.1	0.1	0.1	0.0	0.1
Basophils	0.5–1%	0.1	0.2	0.3	0.2	0.9	0.2	0.2
Large Unstained Cells	<4%	2.4	2.0	2.0	2.9	6.4	3.3	3.6
C Reactive protein	5–10 mg/L	19.1	6.79	10	10.8			
Pro calcitonin	<0.05 ng/mL	0.06	0.05	0.10	0.10			
Erythro sedimentation rate	<28 mm/hours			16	16			
Β-d glucan	<3	<3	<3					

**Table 2 reports-06-00011-t002:** Relative table of concentrations, expressed in Figure 3. A1 CP: Positive resistant control. Red color represents the resistant concentration (see Figure 3), while blue represents color the sensible concentration (see Figure 3).

	1	2	3	4	5	6	7	8	9	10	11	12
A	CP	APH:16	8	4	2	1	0.5	0.25	0.125	0.0625	0.031	ITR 4
B												2
C												1
D	VOR: 8	4	2	1	0.5	0.25	0.125	0.0625	0.031	0.015	0.0078	0.5
E	POS: 8	4	2	1	0.5	0.25	0.125	0.0625	0.031	0.015	0.0078	0.25
F												0.125
G												0.0625
H												0.031

APH Amphotericin B; from A2 to A 7 sensible concentrations; from A8 to A 11 resistant concentrations. VO: Voriconazole: from D1 to D6 sensible concentrations; form D7 to D11 resistant concentrations. Pos: Posaconazole: from E1 to E6 sensible concentrations; form E7 to E11 resistant concentrations. ITR: Itraconazole: A12-F12 sensible concentrations; G12 and H12 resistant concentrations.

**Table 3 reports-06-00011-t003:** Characteristics of the test commonly used for the diagnosis of Cryptococcal infections.

Test	Type of Test	Sensitivity %	Sample	Drawbacks
India ink	Rapid test	30–80	CSF	low sensitivity, technical subjectivity
Culture	Bird seed agar	50–80	CSF, blood	requires extensive laboratory infrastructure; long incubation time (2 days); low sensitivity
Latex agglutination and ELISA Test (Serology)	Early diagnosis in asymptomatic HIV + patients (20–45 min)	90	CSF, serum	expensive, requires extensive laboratory infrastructure
Lateral flow assay (immuno- chromatography)	Early diagnosis in asymptomatic HIV + patients (10 min)	95	CSF, serum	

## Data Availability

Not applicable.

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
