# Peer review of "A Possible New Diagnostic Method for Early Diagnosis of Cryptococcus Infection in Lymphoma Patient Co-Infected with SARS-CoV-2"

_reports, 2023, doi:10.3390/reports6010011_

Round 1

Reviewer 1 Report

Minor corrections are pointed out in the text and that may be corrected before final submission  of article 

Author Response

Dear referee 1 

thank you for your comments that improved our case report. We performed the requested changes.

regarding the question of data before the 14 May, we have not this information because we the patient referred to our ward only in this data and we haven't previous information  

Reviewer 2 Report

Authors present an interesting case of Cryptococcosis diagnosis using the LFA-CrAg test in a Lymphoma patient. However, some aspects need to be addressed:

1. Lane 51- 52, "In the serum of this woman, using a dilution pattern through the CrAg Lateral Flow Assay (Immy, Norman, OK 73069, USA) we detected the antigen of C. neoformans". Please refer how the authors are able to conclude the presence of C. neoformans antigen, since this assay does not discriminate between species. Species was confirmed later by Maldi-Tof, but not from the CrAg test

2. Lanes 58-63: what do the authors refer to "with a high sensitivity to Posaconazole, Amphotericin B and Itraconazole". Please include the MIC values, since there are no breakpoints for Cryptococcus

3. Authors conclude: "We reported the infection of C. neoformans in a woman with non-Hodgkin lymphoma and covid-19. Even if several manuscripts described the development of meningitis sustained by C. neoformans in patients with immunocompromising conditions (9), this is the first case suggesting that an antigen dilution can be used to perform a rapid diagnosis and to quickly start the pharmacological treatment". 

Conclusion must be adjusted since it is well known that the LFA-CrAg assay is used for early detection of cryptococcosis and on-time treatment; therefore, this is not the first case as authors suggest

Author Response

  1. Lane 51- 52, "In the serum of this woman, using a dilution pattern through the CrAg Lateral Flow Assay (Immy, Norman, OK 73069, USA) we detected the antigen of C. neoformans". Please refer how the authors are able to conclude the presence of C. neoformans antigen, since this assay does not discriminate between species. Species was confirmed later by Maldi-Tof, but not from the CrAg test

Answers: it is correct, using antigen test we detected the infection sustained by Cryptococcus, that was identified using Maldi test. We clarify this point

  1. Lanes 58-63: what do the authors refer to "with a high sensitivity to Posaconazole, Amphotericin B and Itraconazole". Please include the MIC values, since there are no breakpoints for Cryptococcus

Answers: as requested MIC Values were added.

  1. Authors conclude: "We reported the infection of C. neoformans in a woman with non-Hodgkin lymphoma and covid-19. Even if several manuscripts described the development of meningitis sustained by C. neoformans in patients with immunocompromising conditions (9), this is the first case suggesting that an antigen dilution can be used to perform a rapid diagnosis and to quickly start the pharmacological treatment". 

Conclusion must be adjusted since it is well known that the LFA-CrAg assay is used for early detection of cryptococcosis and on-time treatment; therefore, this is not the first case as authors suggest

Answer: it is correct, we changed it

Reviewer 3 Report

The authors of the manuscript titled "Cryptococcus infection in Hodgkin’s Lymphoma patient, a possible new diagnostic method for early identification" repost the case of woman with lymphoma no Hodgkin that developed an infection caused by C. neoformans. The body of work presented here is appropriate for the reports; however, it needs minor revisions before it can be considered for publication.

Points that need to be addressed.

  1. The authors should publish this work as a case report rather than an article. Can the authors account for the reciprocity of this method on other patients with similar conditions?.
  2.  If the authors can provide the results of the immunochromatographic assay and mass spectrometry assay as figures, it can improve the quality of the manuscript.
  3. The addition of material and methods will provide the readers with more information regarding all the diagnostic techniques used.
  4. In lines 17 and 56, it should be liposomal amphotericin B not liposomial. Please fix the spelling.
  5.  In Table 1, C reactive protein mg/l should be mg/L.

Author Response

Reviewer 3

Dear Reviewer thank you for your comments that we considered in the revision of the present case report. We changed it in agreement with your suggestions and we also send you a point by point rebuttal letter.

  1. The authors should publish this work as a case report rather than an article. Can the authors account for the reciprocity of this method on other patients with similar conditions?.

Answers: bibliography has been revised

  1.  If the authors can provide the results of the immunochromatographic assay and mass spectrometry assay as figures, it can improve the quality of the manuscript.

Answers: the figures of immunochromatographic assay and mass spectrometry assay have been added

  1. The addition of material and methods will provide the readers with more information regarding all the diagnostic techniques used.

Answers: we have not added the material and methods as a specific section bu we added the figures of the results

  1. In lines 17 and 56, it should be liposomal amphotericin B not liposomial. Please fix the spelling.

Answers: we revised this point according

  1.  In Table 1, C reactive protein mg/l should be mg/L.

Answers: Table 1 has been revised